# Exercise and Quality of Life in Women with Menopausal Symptoms: A Systematic Review and Meta-Analysis of Randomized Controlled Trials

**DOI:** 10.3390/ijerph17197049

**Published:** 2020-09-26

**Authors:** Thi Mai Nguyen, Thi Thanh Toan Do, Tho Nhi Tran, Jin Hee Kim

**Affiliations:** 1Department of Integrative Bioscience & Biotechnology, Sejong University, 209 Neungdong-ro, Gwangjin-gu, Seoul 05006, Korea; mainguyen@sju.ac.kr; 2Institute for Preventive Medicine and Public Health, Hanoi Medical University, 01 Ton That Tung, Dong Da, Hanoi 100000, Vietnam; dothithanhtoan@hmu.edu.vn (T.T.T.D.); tranthonhi@hmu.edu.vn (T.N.T.)

**Keywords:** exercise, quality of life, menopausal symptoms, meta-analysis

## Abstract

Menopausal symptoms are associated with deterioration in physical, mental, and sexual health, lowering women’s quality of life (QoL). Our study objective is to examine the effect of exercise on QoL in women with menopausal symptoms. After initially identifying 1306 studies published on PubMed, Web of Science, Scopus, and Cochrane Library before June 2020, two researchers independently selected nine randomized controlled trials (RCTs) in which any type of exercise was compared with no active treatment. We assessed the risk of bias in the included studies using the Cochrane risk-of-bias 2.0 tool for RCTs and computed the converged standardized mean difference with a 95% confidence interval. We found evidences for the positive effects of exercise on physical and psychological QoL scores in women with menopausal symptoms. However, there was no evidence for the effects of exercise on general, social, and menopause-specific QoL scores. The most common interventions for women with menopausal and urinary symptoms were yoga and pelvic floor muscle training (PFMT), respectively. In our meta-analyses, while yoga significantly improved physical QoL, its effects on general, psychological, sexual, and vasomotor symptoms QoL scores as well as the effect of PFMT on general QoL were not significant. Our findings suggest that well-designed studies are needed to confirm the effect of exercise on QoL in women with menopausal symptoms.

## 1. Introduction

Menopause, which is defined as the permanent cessation of menstrual periods [1] and categorized into premenopausal, perimenopausal, and postmenopausal stages [2], can either occur naturally, generally between the ages of 42 and 58 years [2], or be induced by medical treatments [3]. There is a great variety of menopausal symptoms, of which the four most frequently reported by middle-aged women are vasomotor symptoms (i.e., night sweats and hot flashes), difficulty sleeping/insomnia, vaginal dryness/dyspareunia, and adverse mood/depression [4]. These symptoms may last for years in the postmenopausal period [1]. Existing evidence suggests that as many as 85% of menopausal women experience at least one hot flash, although large differences were found with respect to the frequency, severity, and duration of hot flashes [5]. The reported incidences of vaginal discomfort and sexual dysfunction are 60% [6] and 87% [7], respectively. A significant negative relation between menopausal symptoms and quality of life (QoL) in middle-aged women has been observed in several national [8,9] and multinational studies [10].

Understanding QoL, which is a significant target and concept of practice and research in health and medicine area, is important for relief of symptoms, care, and patient rehabilitation [11]. To alleviate the aforementioned symptoms and increase QoL in women with menopausal symptoms, various therapies, ranging from conventional to complementary and alternative medicine (CAM), have been developed throughout the past decades [12]. Menopausal hormone therapy (MHT) is known to be the most effective treatment for vasomotor symptoms and the genitourinary syndrome of menopause [13], which is a new term describing a group of signs and symptoms including genital (irritation, dryness, and burning) and sexual (impaired function, lack of lubrication, and discomfort or pain), and urinary symptoms (recurrent urinary tract infections, urgency, and dysuria) [14]. However, the Women’s Health Initiative study conducted in 2002 indicated that MHT increases cardiovascular risk, and that, overall, its harm exceeds its benefits [15]. This sparked a great deal of concern over effective non-hormone therapies, including prescription and nonprescription therapies [16,17]. Prescription therapies, prime examples of which are plant-based therapies and herbal medicinal products, have been systematically reviewed; they produce a modest reduction in the frequency of vaginal dryness and hot flashes but no significant reduction in night sweats [18,19]. Nonprescription CAM therapies, including nutritional, physical, psychological, herbal, and folk remedies, are also commonly used, with approximately 80% of women using at least one CAM treatment during a 6-year period [20].

Exercise is an inexpensive intervention that has many significant health benefits. Recently reported randomized controlled trials (RCTs) suggest that exercise positively affects women’s health during the menopausal period [21,22], as well as leading to the prevention of several chronic diseases [23]. A systematic review assessing 28 RCTs with 2646 early postmenopausal women found that, along with a weight-reducing diet, training was likely to preserve normal body weight and bone mineral density as well as to increase muscle strength [21]. However, the association between exercise and menopause-specific QoL remains inconclusive due to inconsistencies among studies [24,25,26,27,28,29,30,31,32]. The aim of this study, therefore, is to systematically review existing evidence and add a quantitative examination of the effect of exercise-based interventions on QoL in women with menopausal symptoms.

## 2. Materials and Methods

This study was reported following the PRISMA guidelines [33].

### 2.1. Search Strategy

Two researchers (T.M.N. and T.N.T.) developed a strategy for searching on the PubMed, Web of Science, Scopus, and Cochrane Library databases to identify relevant studies published up to June 2020. The strategy combined keywords relating to exercise, QoL, and menopausal symptoms (Appendix B, Table A1). For instance, the terms used for PubMed search were as follows: ((“menopause” [Mesh] OR menopause OR “climacteric” [Mesh] OR “hot flashes” [Mesh] OR hot flash OR night sweat OR vasomotor symptom) AND (exercise OR physical activity OR yoga OR tai ji OR qi gong OR aerobic OR sport OR pilate OR movement OR walk OR swim OR run OR train OR dance OR climb) AND (“quality of life” [MESH] OR “quality of life” OR “life quality” OR “value of life” OR “life value” OR “quality of well-being” OR “QoL” OR “HRQoL” OR “HRQL” OR “QWB”) AND (“randomized controlled trial” OR RCT OR controlled clinical trial [PT] OR trial [Tiab] OR randomly [Tiab] OR randomized [Tiab] OR groups [Tiab])). Titles, abstracts, and full texts of the reference lists of review papers and included studies were also screened for additional relevant studies [34].

### 2.2. Selection Criteria

To be relevant, studies had to meet the inclusion and exclusion criteria as follows:(1)Participants: studies of women with at least one menopausal symptom due to the natural decline of reproductive hormones, comprehensive cancer treatment program, hysterectomy, and/or premature ovarian failure were included. Studies of menopausal women that did not mention any menopausal symptoms as inclusion criteria were excluded;(2)Intervention and control groups: studies comparing exercise with no active treatment were included. In the present study, exercise was defined as a planned, structured, repetitive, and purposeful subcategory of physical activity, aimed at the maintenance or improvement of one or more components of physical fitness [35]. There were no restrictions on the frequency and duration of the intervention as well as whether exercises were supervised by instructors or self-delivered. Studies in which exercises were combined with non-exercise methods were excluded because these combined methods possibly influence the actual effect of exercise on QoL in women with menopausal symptoms. We also excluded studies in which the control groups performed active non-exercise interventions such as MHT and cognitive behavior therapy. In addition, we excluded studies where both the intervention and the control groups performed exercises (i.e., aerobics vs. resistance training);(3)Outcomes: QoL scores of both the intervention and the control groups must be provided;(4)Only RCTs of humans published in English were included. Publications in commentary, editorial, or review form were excluded;(5)Studies with unavailable full texts were excluded.

### 2.3. Screening Data

After one researcher (T.M.N.) conducted a search and removed duplicate studies, the titles and abstracts of the remaining studies were screened by two researchers (T.M.N. and T.N.T.), both of whom then screened the full texts of possibly relevant studies in further detail. The whole process was performed independently based on the selection criteria using reference management software (EndNote X9). Disagreements between the findings of these two researchers were discussed with a third researcher (T.T.T.D.) until a consensus was reached.

### 2.4. Data Extraction

Data on the included studies, including the first authors’ names, years of publication, participants’ characteristics, locations, descriptions of intervention and control, QoL questionnaires, and main results, were extracted for qualitative synthesis by one researcher (T.M.N.) and cross-checked by another researcher (T.N.T.). This analysis aimed to compare and assess the included studies in terms of methodological characteristics, interventions used, and reported findings.

Sample sizes and QoL scores of the intervention and the control groups were extracted and divided into general health and menopause-specific QoL domains for meta-analysis by one researcher (T.M.N.) and then cross-checked by another researcher (T.T.T.D.). When a project was repeatedly published from different perspectives, the main report was used as the reference, which was supplemented by other articles.

### 2.5. Risk-of-Bias Assessment

Risk of bias (ROB) was assessed independently by two researchers (T.M.N. and T.N.T.) using the Cochrane ROB 2.0 tool for RCTs, which divides potential bias into five domains, ROBs produced from the randomization process, from deviations developed by the intended interventions, from missing outcome data, from different outcome measurement methods, and from preferable selection of the reported results. This tool also guided the assessment of ROBs overall and by each domain as low ROB, some concerns, or high ROB [36]. Any inconsistencies were resolved by another researcher (T.T.T.D.).

### 2.6. QoL Outcomes

The primary and secondary outcomes of this review are general health and menopause-specific QoL scores, respectively, which were measured by three generic and eleven menopause-specific QoL questionnaires. Based on the relative similarities of questions used in the questionnaires, we grouped the extracted QoL scores into nine domains (Table 1). General health QoL comprises four domains, the general, physical, psychological, and social domains, while menopause-specific QoL included five domains, the vasomotor, sexual, urinary, somatic, and total symptoms domains. Both Greene Climacteric Scale (GCS) and Functional Assessment of Cancer Therapy for Endocrine Subscale (FACT-ES) measured total symptoms by summing the scores on all individual questions about menopausal symptoms used in these questionnaires.

### 2.7. Meta-Analysis

The means and standard deviations (SDs) of QoL scores in the intervention and the control groups after intervention were extracted to determine the effect of exercise on QoL. The extracted data were used to calculate the standardized mean difference (SMD) and its 95% confidence interval (CI). Because of the variety of questions used to assess QoL in the surveyed studies, we calculated SMD instead of mean difference [51]. As the direction of the QoL questionnaires differed in the general, physical, psychological, sexual, and total symptoms QoL domains (Table 1), we multiplied the means of QoL from one set of studies by −1 without any change in SDs [52]. The magnitude of the effect size was defined as small (SMD = 0.2 to 0.5), medium (SMD = 0.5 to 0.8), or large (SMD > 0.8) [53].

Subgroup analyses of the most commonly used interventions were also performed to indicate the effect of each of these interventions on QoL in women with menopausal symptoms.

Heterogeneity among individual studies in each QoL outcome group was assessed by I^2^, χ^2^, and *p*-value. The magnitude of heterogeneity was classified as low heterogeneity (I^2^ = 0% to 24%), moderate (I^2^ = 25% to 49%), substantial (I^2^ = 50% to 74%), or considerable (I^2^ = 75% to 100%) [51].

Publication bias was assessed via funnel plot and Egger’s test. Roughly symmetrical funnel plots were considered to indicate a low publication bias [54]. The test for funnel plot asymmetry was only used when there were at least ten studies included in a meta-analysis [51]. Review Manager Software (RevMan) version 5.3 (ClickTime, San Francisco, CA, USA) was used for all statistical analyses.

## 3. Results

### 3.1. Study Selection

A total of 1306 studies were initially identified. After 352 duplicates were removed, 954 publications were eligible for title and abstract review. Of those, 864 studies were excluded based on the selection criteria (for example, because they used cognitive behavioral therapy, relaxation training, or nutritional intervention). After screening 90 possibly relevant studies as full texts, we further excluded 81 publications including (1) eight non-RCTs; (2) twelve unavailable full texts; (3) four conference abstracts; (4) six non-English articles; (5) five studies combining exercise with other interventions; (6) one study in which both the intervention and the control groups performed exercises; (7) one study using active treatment control group; (8) three studies not providing QoL quantitative outcomes; (9) six studies of menopausal women that did not mention any menopausal symptoms as inclusion criteria; (10) five studies published as different perspectives of the same project with the included studies; (11) 30 studies having a combination of the aforementioned reasons. Detailed reasons for full-text exclusion are presented in Appendix A. Next, of the nine articles included in the qualitative synthesis, the study by Mercier et al. [24] was excluded as it is a single-arm study conducted as a substudy of an RCT. Finally, the eight remaining studies were included in the quantitative synthesis (meta-analysis). No further relevant study was found by checking the reference lists of the nine included studies and review papers. Figure 1 outlines the selection flow.

### 3.2. Characteristics of Included Studies

Table 2 describes the characteristics of the nine relevant studies used for qualitative analyses. The sample size of the included studies ranged from 29 to 254 women. The pooled number of participants in both the intervention and the control groups of the included studies was 882 women with menopausal symptoms. We included only one intervention and one control group taken from each RCT except for the study by Elavsky et al. (2007) [32], which assigned participants to two intervention groups (walking and yoga) and one control group. Therefore, there were ten intervention and nine control groups in this review. Five of nine eligible studies were conducted in the Western Hemisphere (two in Brazil [25,29], two in the USA [28,32], and one in Canada [24]), while four studies were conducted in Asia or Europe (two in Asia [26,27] and two in Europe [30,31]).

The effects of pelvic floor muscle training (PFMT), yoga and yoga-like postures, aerobic training, walking, and self-directed exercise programs (e.g., swimming, running, cycling) were investigated in three [24,25,29], four [26,27,28,32], one [30], one [32], and one [31] study, respectively. All interventions, except aerobic training [30], were supervised by therapists or certified instructors. Unsupervised aerobic training was performed four times per week, including at least two mandatory walking sessions along with jogging, swimming, aerobics, or other gymnastic exercises [30]. The duration of the interventions ranged from 4 weeks to 6 months. Exercise doses varied from 20 to 90 min per bout and from one to seven times per week.

While three studies [24,25,29] on urinary symptoms used only one specific QoL questionnaire (International Consultation on Incontinence Questionnaire (ICIQ) or King’s Health Questionnaire (KHQ)), two studies [28,30] on daily hot flashes used both generic and specific QoL instruments, including visual analogue scale (VAS), 36-Item Short-Form Health Survey (SF-36), Hot-Flash-Related Daily Interference Scale (HFRDIS), and Women’s Health Questionnaire (WHQ). In addition, Duijts et al. (2012) [31] used up to five questionnaires in women with more than two menopausal symptoms.

Because type, frequency, and intensity of exercise intervention, QoL questionnaires, and study populations differed among the included studies, we used a random-effects approach in all meta-analyses.

Regarding the main findings of the included studies, three studies investigating the association between PFMT and QoL in women with urinary symptoms found several positive results [24,25,29]. While the single-arm study by Mercier et al. [24] and the RCT by Bertotto et al. [25] consistently showed statistically significant associations, Pereira et al. [29] showed a statistically significant large effect of PFMT on urinary symptoms and a statistically non-significant small effect of PFMT on general QoL. The effects of yoga, however, were conflicting. While three studies illustrated a positive relation, two of them [26,27] found an extremely high statistically significant improvement in the intervention groups, but Avis et al. [28] suggested that yoga may not offer any advantages over other types of intervention (Table 2). Luoto et al. [30], Elavsky et al. [32], and Duijts et al. [31] found positive effects of aerobic exercise, walking, and an exercise program, respectively, on QoL in women with menopausal symptoms (Table 2).

### 3.3. General Health QoL Outcomes

The pooled effect of exercise on general QoL was positive with a small effect size, but the relation was not statistically significant (SMD = 0.23, 95% CI: −0.1 to 0.56) (Figure 2). Positive associations between exercise and physical, psychological, and social domains were observed in all individual studies despite a wide range of effect sizes. The physical and psychological pooled effect sizes showed some positive effects of exercise on QoL in women with menopausal symptoms (SMD = 0.89, 95% CI: −0.11 to 1.89, *p* = 0.08 and SMD = 0.56, 95% CI: −0.04 to 1.15, *p* = 0.07, respectively) (Figure 2). Although a similar positive association was found in the social domain in the study conducted by Jayabharathi et al. [27] (SMD = 1.61, 95% CI: 1.33 to 1.89), the pooled effect size did not reveal any significant difference between the intervention and the control groups (SMD = 0.94, 95% CI: −0.37 to 2.26) (Figure 2).

### 3.4. Menopause-Specific QoL Outcomes

The meta-analyses did not reveal any significant differences (*p* > 0.05) in QoL relating to menopausal symptoms between women in the exercise and the control groups (Figure 3). Urinary domains showed the largest improvement (SMD = −0.79, 95% CI: −1.92 to 0.34). Based on Cohen’s categorization, the pooled effect sizes of the vasomotor, sexual, somatic, and total symptoms domains were all small (SMD = −0.14, 95% CI: −0.42 to 0.15, SMD = −0.19, 95% CI: −0.43 to 0.04, SMD = −0.02; 95% CI: −0.23 to 0.19, and SMD = −0.18, 95% CI: −0.39 to 0.03, respectively) (Figure 3).

### 3.5. Subgroup Analyses

We found a non-significant improvement in general QoL in women with urinary symptoms after PFMT interventions compared with no intervention (SMD = 0.76, 95% CI: −0.40 to 1.92, *p* = 0.20). In this review, we could not confirm the effect of yoga on general QoL (SMD = −0.07, 95% CI: −0.46 to 0.33). However, there were evidences for positive associations between yoga and physical, psychological, sexual, and vasomotor symptoms QoL scores. Of these, the physical domain showed the strongest association, with a statistically significant difference between the intervention and the control groups (SMD = 1.39, 95% CI: 0.19 to 2.59, *p* = 0.02) (Table 3). Although the effect size of the psychological domain was nearly large (SMD = 0.76; 95% CI: −0.3 to 1.81), the two groups did not significantly differ (*p* = 0.16).

### 3.6. Risk-of-Bias Assessment

Table 4 summarizes the ROB assessment. Except the study by Luoto et al. [30], judgements about the overall ROB of eight remaining RCTs express some concerns. The ROBs relating to selection of the reported result and randomization process were low in all and in most of the included studies, respectively. Despite mentioning a random assignment of participants to exercise and control groups, two studies [26,27] insufficiently described the process. Despite relatively high dropout rates (Table 2), five [25,26,28,30,32] out of nine studies did not provide adequate evidences that the results were not biased by missing outcome data, reporting some concerns about this ROB domain. In general, the ROBs due to deviations from intended interventions and measurement of the outcome were considerable.

### 3.7. Publication Bias

Because all meta-analyses of the effects of exercise of QoL domains included fewer than ten studies, the test for funnel plot asymmetry was not performed. Figure 4 shows several roughly symmetrical funnel plots on general, physical, psychological, vasomotor symptoms, and sexual symptoms domains, indicating a low risk of publication bias. As the number of studies is low in many analyses, the asymmetry assessment for the publication bias was difficult.

## 4. Discussion

The present study provided evidences for positive effects of exercise on physical and psychological QoL scores in women with menopausal symptoms. However, there was no evidence for the effects of exercise on general, social, and menopause-specific QoL scores. Yoga and PFMT, respectively, were the most common interventions for women with menopausal and urinary symptoms in the studies included in this review. In our meta-analyses, while yoga significantly improved physical QoL, its effects on general, psychological, sexual, and vasomotor symptoms QoL scores as well as the effect of PFMT on general QoL were not significant.

To the best of our knowledge, there have not been any meta-analytical studies examining the effect of exercise on either general or menopause-specific QoL in women with menopausal symptoms with which our findings may be compared. The majority of previous systematic reviews focused on menopausal symptoms in terms of frequency and intensity [21,55,56,57]. A recently published meta-analysis only identified an association between physical exercise and quality of life related to menopausal sexual symptoms [58].

While most women experience menopause naturally between the ages of 42 and 58 [2], some women experience menopause earlier as a result of either surgical intervention or damage to the ovaries [3]. Several symptoms of menopause profoundly affect women’s health throughout the remainder of their life [1]. Although not all women experience menopausal symptoms, most women (about 85%) experience bothersome menopausal symptoms as well as complications thereof [59]. For this reason, we focused on women with menopausal symptoms to investigate the negative impact of menopause on women’s life. Furthermore, an article that systematically reviewed qualitative evidence on women’s experience of menopause found that women experience menopause in different ways depending on their personal, familial, and sociocultural background, and the experience could be either positive or negative [60]. Therefore, QoL, defined as “an individual’s perception of their position in life, in the context of the culture and value system in which they live, and in relation to their goals, expectations, standards, and concerns” [61], could be used to indicate the extent to which menopause affects a woman’s life. For this reason, the present study solely focused on QoL in women with menopausal symptoms. This is one major difference between our study and previous trials of menopausal women that did not mention any symptoms. Nevertheless, all studies including ours have consistently indicated that there are some differences between the intervention and the control groups in terms of physical and psychological QoL domains [62,63,64,65].

Several review articles about women’s health during the menopausal transition or early postmenopausal stage suggested considerable positive effect of exercise on health-related fitness [66], which is directly associated with QoL [67]. In particular, Asikainen et al. [21] showed that early postmenopausal women could benefit from moderate walking everyday along with weekly resistance training in terms of morphological fitness (measured by bone strength, weight, and body fat), musculoskeletal fitness (measured by various muscle tests and flexibility tests), motor fitness (measured by balance and coordination tests), cardiorespiratory fitness (measured by maximal oxygen consumption and blood pressure), and metabolic fitness (measured by blood levels of glucose, low-density lipoprotein, high-density lipoprotein, triglycerides, total cholesterol, and insulin). A later review by Sternfeld et al. [22] found similar results as well as improvements in bodily pain, physical function, and mental health in active women. In other words, exercise positively affects general health-related QoL in women with menopausal symptoms, especially the physical and psychological QoL domains. This is in line with our finding that many intervention–control pairs included in the meta-analyses showed improvement in the four general health QoL domains among exercising women compared with the control group, although different studies showed different magnitudes of improvement. Avis et al. [28] showed a non-significant negative association between yoga and general QoL, but they reported several limitations, including an underpowered sample size, a challenging participant recruitment process, and poor class attendance because of the inability to offer more flexible and varied intervention class days and times. Providing more options for classes, instead of only offering the class once per week, would improve both recruitment and class attendance.

The effects of exercise on women’s health vary widely, depending in part on the menopausal symptoms that are assessed [22]. A previous review by Atapattu et al. [66] provided several plausible mechanisms to explain the effect of exercise on vasomotor symptoms, such as increased vagal tone, the influence of stress hormones and parasympathetic activation, and thermoregulatory center activity. This study suggests that exercise may improve vasomotor symptoms. However, previous studies found insufficient evidence to determine the effects of exercise on hot flashes and night sweats [22,55,57,66]. These results were similar to our findings in that there was no significant association between exercise and vasomotor-symptom-related QoL. A similar pattern was found concerning sexual symptoms. There was substantial evidence showing the beneficial effects of exercise on oxytocin, cortisol, and estrogen levels, all of which are hormones that affect sexual function and orgasms. However, the association between exercise and sexual symptoms or QoL domains remains inconclusive [58]. Although regular exercise is theoretically believed to help improve menopausal symptoms and related QoL domains, current evidence of its beneficial effect is generally inconsistent. This may be due in part to differences in study designs, study populations, inclusion criteria, interventions, and outcome measures between the included studies. In the somatic and total symptoms QoL domains, inconsistencies among effect sizes could result from the small number of intervention–control pairs included in the meta-analyses. In the same manner, the only specific QoL domain that showed a large effect size with a consistent direction of associations in the included studies was urinary symptoms. The non-significant difference between the exercise and the control groups could be due to the high degree of heterogeneity among the selected studies (*p* < 0.05).

Regarding different types of exercise interventions, there were three studies [24,25,29] investigating the effect of PFMT on QoL in women with urinary symptoms included in this review. Although the study by Mercier et al. [24] was a single-arm study, it was not excluded from the qualitative synthesis because this study was a substudy of an RCT [68] which included three parallel evaluations at pre-intervention, 1–2 weeks after interventions, and a year post randomization, respectively. Data reported in the included study [24] were extracted from the first two evaluations. As the mother study [68] was strictly designed as an RCT, the study by Mercier et al. [24] was considered to meet the related selection criterion in this review. Furthermore, because the aims of the qualitative analysis were to compare and to assess the included studies in terms of methodological characteristics, interventions used, and reported findings, we included this study to provide a broad overview of the association between PFMT and QoL in women with urinary symptoms.

Similar to the main findings of three included studies using PFMT, Cacciari et al. [69] indicated that PFMT can reduce the number of urinary leakage episodes and improve all types of urinary incontinence issues as well as QoL in women with urinary symptoms. Furthermore, Carcelen-Fraile et al. [58] found that PFMT was the most common type of exercise used in studies investigating the effects of physical exercise programs on quality of sexual life related to menopausal symptoms, and it seems most beneficial for sexual function in perimenopausal or postmenopausal women. These findings suggested that PFMT should be prescribed as a conservative therapy for women with urinary symptoms. However, a non-significant improvement revealed in our subgroup analysis using a random-effects model of the effect of PFMT on general QoL in women with urinary symptoms highlighted a need for more supporting evidences for the aforementioned suggestion. In our additional fixed-effect model analyses, we found a significant association between PFMT and general QoL (SMD = 0.70, 95% CI: 0.16 to 1.24). Nevertheless, its usage seems debatable because of an unexplained considerable heterogeneity (I^2^ = 78%). Besides, although two studies [25,29] included in such analysis both recruited Brazilian women with urinary symptoms, used two supervised PFMTs per week as the intervention compared with no treatment and used a similar question to measure general QoL, the true intervention effects were rarely similar in studies conducted independently. 

Additionally, a systematic review of RCTs found that 30 min of walking every day combined with resistance training twice a week has various health benefits for early postmenopausal women, and it was recommended that sedentary women start with walking by incorporating it into everyday life [21]. There was no study included in the present review that mixed two different types of exercise. However, three individual studies consistently indicated positive effects of walking, aerobic training, and self-directed exercise programs (e.g., swimming, running, cycling) on QoL in women with menopausal symptoms. 

On the other hand, in this study, we found an inconclusive association between yoga, the most commonly used exercise intervention in the included studies, and QoL in women with menopausal symptoms. This could be because one [28] out of the three studies with a yoga intervention was a pilot study. Moreover, various styles of yoga were used in the included studies, each of which has different characteristics in terms of frequency, intensity, and exercise conditions. Nevertheless, it is worth noting that the yoga subgroup analysis found positive effects of yoga on physical, psychological, sexual, and vasomotor symptoms QoL domains despite different magnitudes of effect sizes. This is partially in line with findings of the study by Cramer et al. [55], reporting that, in comparison with no treatment, yoga reduced total menopausal, psychological, somatic, vasomotor, and urogenital symptoms. Dealing with potential methodological biases in the included studies, both reviews suggested that more well-designed studies are needed to confirm the effect of yoga on health status of women with menopausal symptoms.

To the best of our knowledge, this is the first meta-analytic review investigating the association between exercise and QoL in women with menopausal symptoms. To minimize the methodological heterogeneity among the included studies, we only selected RCTs, which are designed to minimize the risk of bias, and are considered as the gold standard for evidence-based medicine. To confirm the effect of exercise, we only selected studies in which exercise was compared with no active treatment. In most cases, the control group was waitlist or no intervention. However, it is acceptable that Ngowsiri et al. [26] and Luoto et al. [30] provided both groups with a Menopausal Health promotion handbook and lectures once or twice per month, respectively in order to reduce the dropout rate of the control group without influencing the outcomes. Choosing only women who experienced inconvenience because of menopausal symptoms could reflect the actual effect of exercise on menopause-specific QoL.

The present study also has some limitations. First, some of the included RCTs had small sample sizes. Second, the ROB of the included studies was considerable. This results from the fact that blinding of participants, people delivering the interventions, and outcome assessors in studies using exercise interventions is almost impossible; besides, most QoL questionnaires used were self-reported. Third, the substantial and considerable heterogeneity in most of meta-analyses probably affects the interpretability of the results. Next, it is difficult to evaluate publication bias when the number of studies included in meta-analyses are low. Finally, although a comprehensive search was performed, selection biases, which are potential threats to all systematic reviews, may exist.

## 5. Conclusions

This systematic review found positive effects of exercise on physical and psychological QoL scores in women with menopausal symptoms. However, there were no evident effects of exercise on general, social, and menopause-specific QoL scores in women after exercise interventions compared with no active interventions. Yoga and PFMT, respectively, were the most common interventions for women with menopausal and urinary symptoms in the included studies. In our meta-analyses, while yoga significantly improved physical QoL, its effects on general, psychological, sexual, and vasomotor symptoms QoL scores as well as the effect of PFMT on general QoL were not significant. Our findings suggest that well-designed trials are needed to confirm the effect of exercise on QoL in women with menopausal symptoms.

## Figures and Tables

**Figure 1 ijerph-17-07049-f001:**
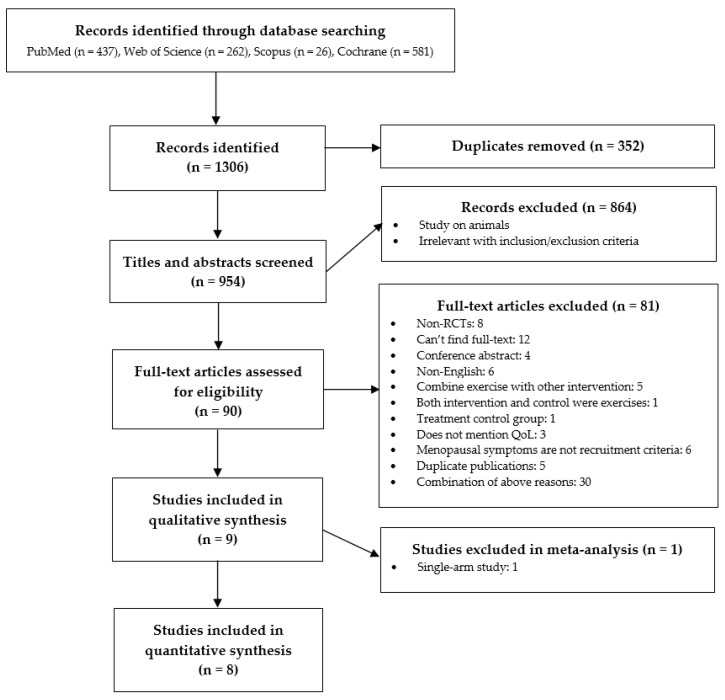
Flow chart of the study selection process.

**Figure 2 ijerph-17-07049-f002:**
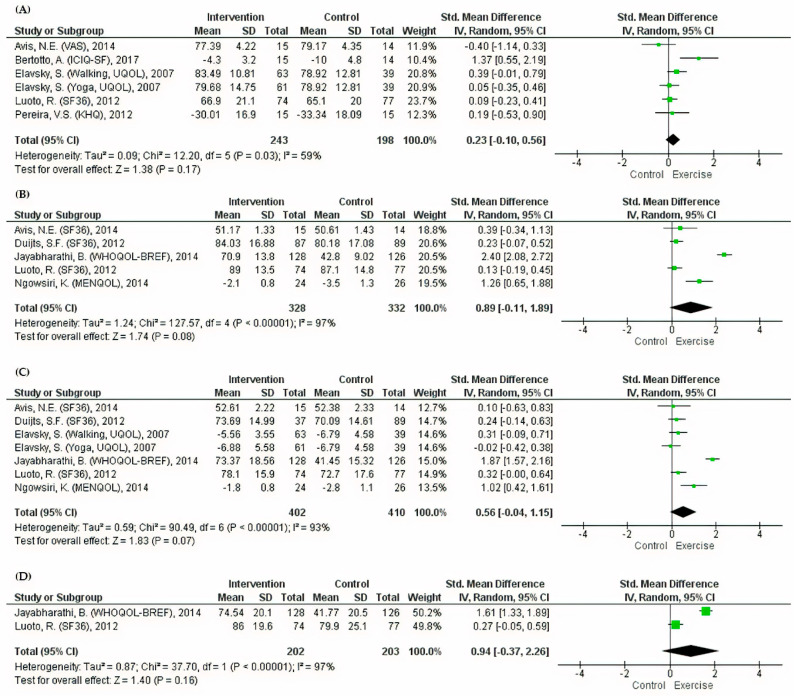
Meta-analyses of the effects of exercise on general QoL domains. (**A**) Forest plot of six datasets on general QoL. (**B**) Forest plot of five datasets on the physical component of QoL. (**C**) Forest plot of seven datasets on the psychological component of QoL. (**D**) Forest plot of two datasets on the social component of QoL.

**Figure 3 ijerph-17-07049-f003:**
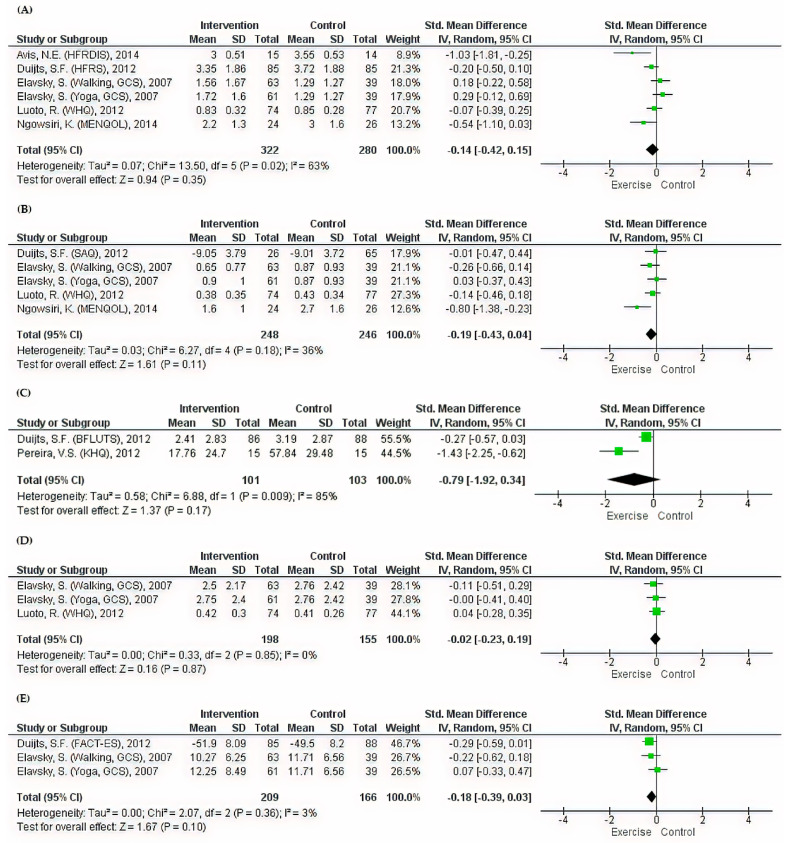
Meta-analyses of the effects of exercise on menopause-specific QoL domains. (**A**) Forest plot of six datasets on vasomotor symptoms. (**B**) Forest plot of five datasets on sexual symptoms. (**C**) Forest plot of two datasets on urinary symptoms. (**D**) Forest plot of three datasets on somatic symptoms. (**E**) Forest plot of three datasets on total symptoms.

**Figure 4 ijerph-17-07049-f004:**
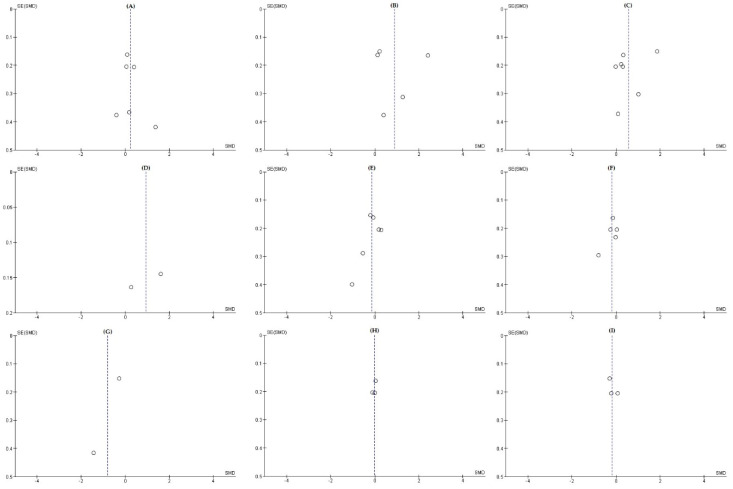
Funnel plots in the meta-analyses of the effects of exercise on nine quality of life (QoL) domains. (**A**) General. (**B**) Physical. (**C**) Psychological. (**D**) Social. (**E**) Vasomotor symptoms. (**F**) Sexual symptoms. (**G**) Urinary symptoms. (**H**) Somatic symptoms. (**I**) Total symptoms.

**Table 1 ijerph-17-07049-t001:** Quality-of-life (QoL) questionnaires and domains.

QoL Questionnaire	Abbreviation	Type ^2^	QoL Domain ^3^								
			General Health				Menopause-Specific				
			General	Physical	Psychological	Social	Vasomotor	Sexual	Urinary	Somatic	Total Symptoms
Global QoL [37]	GQOL/VAS ^1^	G	P	-	-	-	-	-	-	-	-
36-Item Short-Form Health Survey [38]	SF-36	G	P	P	P	P	-	-	-	-	-
World Health Organization QoL—Brief Version [39]	WHOQOL-BREF	G	-	P	P	P	-	-	-	-	-
Utian QoL [40]	UQOL	S	P	-	N	-	-	-	-	-	-
Menopause-Specific QoL [41]	MENQOL	S	-	N	N	-	N	N	-	-	-
International Consultation on Incontinence Questionnaire (Short Form) [42]	ICIQ-SF	S	N	-	-	-	-	-	-	-	-
King’s Health Questionnaire [43]	KHQ	S	N	-	-	-	-	-	N	-	-
Sexual Activity Questionnaire [44]	SAQ	S	-	-	-	-	-	P	-	-	-
Greene Climacteric Scale [45]	GCS	S	-	-	-	-	N	N	-	N	N
Women’s Health Questionnaire [46]	WHQ	S	-	-	-	-	N	N	-	N	-
Hot-Flash-Related Daily Interference Scale [47]	HFRDIS	S	-	-	-	-	N	-	-	-	-
Hot Flash Rating Scale [48]	HFRS	S	-	-	-	-	N	-	-	-	-
Bristol Female Lower Urinary Tract Symptoms [49]	BFLUTS	S	-	-	-	-	-	-	N	-	-
Functional Assessment of Cancer Therapy for Endocrine Subscale [50]	FACT-ES	S	-	-	-	-	-	-	-	-	P

QoL = quality of life; ^1^ As being assessed via a single-item 100 mm visual analogue scale (VAS), GQOL was also called VAS; ^2^ G = generic QoL questionnaire; S = menopause-specific QoL questionnaire; ^3^ P = positive direction (higher score denoted better QoL); N = negative direction (higher score denoted worse QoL).

**Table 2 ijerph-17-07049-t002:** Characteristics of the nine studies included in the qualitative synthesis.

Study	Participant	Country	Dropout Rate	Intervention	Frequency ^1^	Duration ^1^	Control	QoL Questionnaire	Result
Mercier et al. [24]	32 women with GSM; ≥55 years old	Canada	9.4% (3/32)	PFMT; Supervised and home-based (*n* = 32)	Supervised: 1 b/w, 60 min/b Home-based: 5 d/w	12 w	None (*n* = 0)	ICIQ-VS	QoL and sexual function of women with GSM improved after the intervention.
Bertotto et al. [25]	49 postmenopausal women with urinary symptoms; 50–65 years old	Brazil	8.2% (4/49)	PFMT: contraction (*n* = 15)	2 b/w, 20 min/b	4 w	No treatment (*n* = 14)	ICIQ-SF	The PFMT group exhibited significant increases in ICIQ-SF scores.
Ngowsiri et al. [26]	54 menopausal women with menopausal symptoms; 45–59 years old	Thailand	7.4% (4/54)	Rusie Dutton dance of 16 yoga-like postures; Supervised (*n* = 24)	3 b/w, 90 min/b	13 w	Provided a handbook (*n* = 26)	MENQOL	There was a significant improvement in all MENQOL domains in the experiment group and between the two groups.
Jayabharathi et al. [27]	260 women with menopausal symptoms; 45–55 years old	India	2.3% (6/260)	Yoga; Supervised and home-based (*n* = 128)	Supervised: 5 consecutive days: 2 b/d, ~45 min/b; Later: 2 d/w Home-based: 35–40 min/d	18 w	No intervention (*n* = 126)	WHOQOL-BREF	A statistically significant difference between the study group and the control group was observed in terms of all domains of QoL.
Avis et al. [28]	54 menopausal women with ≥4 hot flashes/day; 45–58 years old	USA	20.4% (11/54)	Integral yoga Supervised and home-based qualified DVD for self-practice (*n* = 15)	Supervised: 1 b/w, 90 min/b Home-based: 3 b/w, 15 min/b	10 w	Waitlist (*n* = 14)	SF-36 HFRDIS VAS	Yoga can act as a behavioral option which helps in reducing hot flashes. There was no advantage of yoga over other types of exercise.
Pereira et al. [29]	45 postmenopausal women with urinary symptoms; 53–73 years old	Brazil	8.9% (4/45)	PFMT: contraction; Supervised by a physical therapist (*n* = 15)	2 b/w, 40 min/b	6 w	No intervention (*n* = 15)	KHQ	Several positive results of PFMT in treatment for urinary leakage, pelvic floor muscle pressure, and QoL were observed.
Luoto et al. [30]	176 menopausal women with daily hot flashes; 40–63 years old	Finland	12.5% (22/176)	Unsupervised aerobic training (*n* = 74)	4 b/w, 50 min/b	24 w	1–2 lectures/ month; 60–75 min (*n* = 80)	SF-36 WHQ	Women in the intervention group had significantly higher SF-36 scores in mental health than those in the control group.
Duijts et al. [31]	422 breast cancer patients with menopause symptoms; 48.2 ± 5.6 years old	The Netherlands	16.6% (70/422)	Home-based, self-directed exercise program; Assisted by a physiotherapist (*n* = 104)	Home-based: 150–180 min/w	12 w	Waitlist (*n* = 103)	SF-36 HFRS SAQ FACT-ES BFLUTS	There were significant differences in improvement for menopause symptoms, SAQ, and SF-36 between the intervention and the control group.
Elavsky et al. [32]	164 sedentary women with menopausal symptoms; 42–58 years old	USA	24.4% (40/164)	(1) Walking: supervised; (*n* = 63) (2) Iyengar yoga: supervised (*n* = 62)	(1) 3 b/w; 60 min/b; (2) 2 b/w; 90 min/b	16 w	Waitlist (*n* = 39)	UQOL GCS	The yoga and walking interventions showed positive effects on menopause-specific QoL.

GSM = genitourinary syndrome of menopause; PFMT = pelvic floor muscle training; QoL = quality of life; *n* = number of participants; ICIQ-VS = International Consultation Incontinence Questionnaire—Vaginal Symptoms; ICIQ-SF = International Consultation on Incontinence Questionnaire (Short Form); MENQOL = Menopause-Specific Quality of Life; WHOQOL-BREF = World Health Organization QoL—Brief Version; SF-36 = 36-Item Short-Form Health Survey; HFRDIS = Hot-Flash-Related Daily Interference Scale; VAS = Visual Analogue Scale; KHQ = King’s Health Questionnaire; WHQ = Women’s Health Questionnaire; HFRS = Hot Flash Rating Scale; SAQ = Sexual Activity Questionnaire; FACT-ES = Functional Assessment of Cancer Therapy for Endocrine Subscale; BFLUTS = Bristol Female Lower Urinary Tract Symptoms; UQOL = Utian Quality of Life; GCS = Greene Climacteric Scale; ^1^ b = bout; d = day; w = week; min = minute.

**Table 3 ijerph-17-07049-t003:** Meta-analyses of the effects of PFMT and yoga on QoL domains.

QoL Domain	No. Studies	No. Participants (Exercise)	No. Participants (Control)	SMD (95% CI)	*p*-Value	Heterogeneity ^1^ I^2^; χ^2^; *p*-Value
**PFMT**						
General	2	30	29	0.76 (−0.40 to 1.92)	0.20	78%; 4.52; 0.03
**Yoga**						
General	2	76	53	−0.07 (−0.46 to 0.33)	0.74	12%; 1.14; 0.29
Physical	3	167	166	1.39 (0.19 to 2.59)	0.02	93%; 29.31; *p* *
Psychological	4	228	205	0.76 (−0.3 to 1.81)	0.16	95%; 63.01; *p* *
Sexual	2	85	65	−0.36 (−1.18 to 0.46)	0.39	81%; 5.39; 0.02
Vasomotor	3	100	79	−0.37 (−1.15 to 0.4)	0.34	82%; 11.18; *p* **

QoL = quality of life; PFMT = pelvic floor muscle training; SMD = standardized mean difference; CI = confidence interval; ^1^ * *p* < 0.00001; ** *p* < 0.01.

**Table 4 ijerph-17-07049-t004:** Risk-of-bias assessment of the included studies.

Study	RandomIzation Process	Deviations from Intended Interventions	Missing Outcome Data	Measurement of the Outcome	Selection of the Reported Result	Overall
Mercier et al. [24]	Low	Some concerns	Low	Low	Low	Some concerns
Bertotto et al. [25]	Low	Some concerns	Some concerns	Some concerns	Low	Some concerns
Ngowsiri et al. [26]	Some concerns	Some concerns	Some concerns	Some concerns	Low	Some concerns
Jayabharathi et al. [27]	Low	Some concerns	Low	Some concerns	Low	Some concerns
Avis et al. [28]	Some concerns	Some concerns	Some concerns	Some concerns	Low	Some concerns
Pereira et al. [29]	Low	Some concerns	Low	Some concerns	Low	Some concerns
Luoto et al. [30]	Low	High	Some concerns	Some concerns	Low	High
Duijts et al. [31]	Low	Some concerns	Low	Some concerns	Low	Some concerns
Elavsky et al. [32]	Low	Some concerns	Some concerns	Low	Low	Some concerns

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
