# Peer review of "Exercise and Quality of Life in Women with Menopausal Symptoms: A Systematic Review and Meta-Analysis of Randomized Controlled Trials"

_ijerph, 2020, doi:10.3390/ijerph17197049_

Round 1
Reviewer 1 Report
Corrections and concerns:
- 1, suggest moving figure to the results section, replace the runner with something more age-appropriate for women experiencing menopause
p.3 line 106, do you mean “conducted” instead of “conduced”
- 3 line 126, missing a comma after “reporting bias”
- 4 Table 1, expand QoL domain headings
p.5 lines 156-157 30% is repeated for I2
- 5 Table 2S for the number and why full texts were excluded is missing (from the editors)
- 6 line 192, “gymnastics”-this is not something that menopausal women typically do, please explain
- 7 Table 2, providing minutes of exercise would be better than number of bouts
- 7 Table 2, results are inconsistently reported
- 8, start of section 3.5, aren’t the results statistically significant p=0.01?
- 9, Table 4, there is no explanation for “other bias” despite the fact that the category was unanimously “high”
- 14, conclusion should recommend exercise that is specific to menopause, i.e. B.K.S. Iyengar has yoga practices specifically scripted for menopausal women. Consult the text “B.K.S. Iyengar Yoga: The Path to Holistic Health”
Author Response
Thanks for your valuable comments. We attached our responses to your comments with our revised manuscript. Please check the attached files.

Reviewer 2 Report
Thank you for the opportunity to read this well written, clear and concise study. It was interesting and a pleasure to read. However, a series of considerations are recommended to the authors:
Has this systematic review been included in the prospective international registry of systematic reviews (PROSPERO)?
Line 27: Please add a short explanatory title and caption to the figure.
Line 175: Please explain the legend in figure 1 in more detail.
In section 3.2. Characteristics of included studies: should include the overall dropout rate for all articles and whether they reflect potential adverse effects.
Lines 189-191: Please add the bibliographic citation of the article mentioned.
Line 214: Please, in Table 2 the number of participants should be added, both in the intervention group and in the control group column. Also, in the participant column, you need to add the total number of participants.
Line 217: in the table footer you must add the full name of all the abbreviations used in the table.
Table 3: in the table footer you must add the full name of all the abbreviations used in the table.
Author Response

(The authors gave the same response as above.)

Reviewer 3 Report
In their meta-analysis, the researchers investigate whether non-specified exercise training improves the quality of life in women experiencing menopausal symptoms compared with non-active control. The topic is important as many women live a large portion of their life in the postmenopausal state, and some experience menopausal symptoms for an extended period. As menopausal hormone therapy is not suitable for everyone, investigation of non-pharmacological interventions is warranted. Moreover, quality of life is an important and patient-centred outcome. Thus, the aim of the meta-analysis is justified, and the topic is attractive to many stakeholders from middle-aged women to clinicians and researchers. The authors reason their work clearly and present the study question and study selection criteria according to PICOS. The search strategy comes through as comprehensive, and the search process is described in detail. Therefore, I trust that the authors have found all the relevant literature. In the future, I recommend the authors to pre-register their review protocol for transparency. However, I have some concerns about how the study results are interpreted. The researchers should also weigh the certainty of the evidence and address the study’s limitations more thoroughly.
Broad comments:
In most analyses, the authors have rightfully used random-effects models as included studies are clinically heterogeneous. However, in PFMT subgroup analysis, a fixed-effects model is used. The authors justify the use based on similar study populations, interventions, and outcome measures. Still, statistical heterogeneity was considerable (I2 = 78%). (Though the estimation of heterogeneity is difficult with this study number and I2 should not be solely used to select the appropriate model.) Moreover, in exercise studies, the true intervention effect is rarely similar in different studies. Taken these points into account, the use of fixed-effects model seems debatable. As I repeated the analysis with a random-effects model, the result did not remain statistically significant, although the now wider confidence interval still leans towards PFMT benefits. I recommend the authors to reconsider their model choice or discuss this remaining unexplained statistical heterogeneity as a limitation.
When interpreting the results, the authors use wordings such as “Positive associations between exercise and physical, psychological, and social domains were observed in all individual studies despite a wide range of effect sizes.”, “The physical and psychological pooled effect sizes showed positive effects of exercise on QoL in women with menopausal symptoms, with marginal significances” or “Urinary domains showed the largest improvement” while the results did not show statistically significant associations or effects. Thus, we cannot say for certain that the associations/effects were indeed positive or showed improvement. The data have many uncertainties, and I recommend that the authors consider this while interpreting the results (in the abstract, results, discussion, and conclusions).
The risk of bias assessment was performed according to the older Cochrane RoB tool. The authors determined the risk of other bias being high in all studies. I recommend that the authors explain their judgement briefly in the text as this would be interesting to the readers. I also suggest that the authors provide an overall risk of bias assessment across the studies. At a glance, the risk of bias appears to be significant, and it makes it challenging to fully trust the results obtained from the meta-analysis. This should be addressed in the discussion.
What I find the manuscript is missing, is the certainty of evidence evaluation. For this, the authors could take some inspiration from the GRADE approach. For example, the risk of bias seems to be high, and there appears to be a substantial amount of clinical and statistical heterogeneity. Therefore, I find that the certainty of the evidence favouring exercise is low. This finding is a result in itself, and it highlights the need for more studies looking at the possible quality of life benefits of exercise in menopausal women. The examination of certainty of evidence would heighten the value of the manuscript.
Specific comments:
Abstract
Page 1, lines 19 onward: I recommend presenting results from primary and secondary analyses before presenting the results from the subgroup analyses. Consider how you are interpreting the results.
Page 1, lines 24–25: In my opinion, benefits of PFMT to improve quality of life are still uncertain based on this meta-analysis. Thus, I dont think that any definite recommendations can be made.
Page 1, lines 25-26: I think that the need for well-designed studies is universal across different exercise modalities - not just yoga.
Graphical abstract: I recommend writing confidence intervals as “95% CI x.xx to x.xx” so they are easier to read.
Introduction
I like the structure of the introduction.
Page 2, line 47: Hormone replacement therapy is an oldish term, consider using menopausal hormone therapy or hormone therapy.
Materials and methods
Page 2, lines 73–74: I understand the use of PRISMA, but it is a reporting guideline, not a method tool per se.
Page 2, lines 98: Nonrelevant at this point, but a comparison between exercise and active non-exercise interventions would be interesting.
Page 2, line 102: I understand only selecting RCTs. I recommend stating your reason also here as this is a criterion when assessing the risk of bias in systematic reviews and currently the reasoning gets lost as it is in the discussion part.
Results
Page 5, line 170: The study by Mercier et al. (2019) does not appear to meet the inclusion criteria set by the authors as it is a single-arm study embed in a RCT.
Page 5, Figure 1: I recommend using a more descriptive figure title.
Page 6, 206: Increase in QoL was not significant in the study by Perreira et al. (2012).
(Line numbers end here, also page numbering changes)
Page 2: Can you really say that yoga negatively affected general QoL when SMD is -0.07 (95% CI -0.46 to 0.33)?
Page 3: 3.7. Publication bias: As the number of studies is low in many analyses, the asymmetry assessment is difficult. Perhaps state this.
Discussion (no line numbers here so I provide citations for clearance)
As mentioned above, I think the discussion needs work, and the authors need to be careful about how they interpret the data as there are many uncertainties. For example, consider revising statements such as:
Page 4: “The present study found marginally significant improvements…”
Page 5: “Nevertheless, all studies including ours have consistently indicated that there are statistically significant differences…”
Page 5: “This is in line with our finding that almost all intervention control pairs included in the meta-analyses showed improvement…”
Page 5: “Our literature review provided several plausible mechanisms to explain the effects of exercise…”: I would consider rephrasing this as the researchers’ study question here did not aim to explain mechanisms. Or was this a reference to the study by Asgari et al. (2015)?
Page 6: “Similarly, Cramer et al. (2012) reported that…”: Cramer et al. have redone the meta-analysis recently. Please read “Cramer H, Peng W, Lauche R. Yoga for menopausal symptoms-A systematic review and meta-analysis. Maturitas. 2018, 109, 13-25.” The evidence has built up after the 2012 analysis.
Page 6: The limitations section is very brief. I recommend exploring the risk of bias, heterogeneity and its sources, and certainty of evidence.
Author Response

(The authors gave the same response as above.)

Reviewer 4 Report
Exercise and quality of life in women with menopausal symptoms: a systematic review and meta-analysis of randomized controlled trials
General comments:
This review study examines the effects of exercise on quality of life (QoL) in women with menopausal symptoms. Menopausal is another important phase in the life of a woman, signalling symptoms affecting the physical, mental, and sexual health, which is not properly handle, affecting negatively the QoL. The review is conducted with rigour, methodologically and scientifically sound. There is dearth of information meta-analytical studies examining the effects of exercise on either general or menopause-specific QoL in women with menopausal symptoms. The findings inform evidence-based on exercise and QoL in gynaecological practice using exercise as a pharmacological tool to improve QoL of menopausal women.
I have few queries as follows:
Specific comments:
Line 27: I do not understand the relevance of the Figure in the abstract.
Line 134: ‘while menopause-specific QoL is composed’ The word ‘composed’ is not suitable. I suggest you substitute it to ‘included’.
Line 144: Start the sentence straight with ‘The means….Delete the ‘Numbers of participants’.
Line 208: ‘Avis et al. (2014) [28]’. Remove the 2014 after Avis et al. to read: Avis et al. [28]
Line 209: Luoto et al. (2012) [30]. See comment above
Line 210: Elavsky et al. (2007) [32], and Duijts et al. (2012) [31]. See comment above. Remove the year after the authors.
Line 214: It don’t think it is necessary putting the years in the Table 2. There should be consistency in the numbering style of the journal
Page 1of 18: ‘Although a similar positive association was found in the social domain in the study conducted by by Jayabharathi et al. (2014) [27]’. Delete the year.
Page 2 of 18: ‘Of those,’ Check this again, shouldn’t it be ‘Of these’?
Page 3 of 18: ‘Only Duijts et al. (2012) [31] reported a high risk of selective reporting’. Remove the year after Duijts et al.
Page 3 of 18: Remove the years in Table 4.
Discussion
Page 5 of 18: ‘Asikainen et al. (2004)’. Delete the year and insert the number reference
Page 5 of 18: ‘A later review by Sternfeld et al. (2011)’ . See previous comment above.
Page 5 of 18: ‘Avis et al. (2014)’ See comment above.
Page 5 of 18: ‘There was a great deal of evidence showing’ I would suggest you rephrase the phrase to read ‘There was a substantial evidence showing…
Page 5 of 18: Cacciari et al. (2019). Delete the year and put the appropriate reference number.
Page 5 of 18: ‘Carcelen-Fraile et al. (2020)’ . See my comment above.
Page 6 of 18: ‘Similarly, Cramer et al. (2012)’. See comment above.
Author Response

(The authors gave the same response as above.)

Round 2
Reviewer 3 Report
The authors have clearly put in a lot of effort to improve the manuscript, and I thank them for their detailed responses. I think that the introduction and methods sections are ready; however, I would still recommend that the authors check how they interpret their results.
Broad comments
What I see in the results is uncertainty and wide confidence intervals, which originates from a small number of studies with small sample sizes. Interpreting results from low-powered analyses is difficult. However, absence of evidence is not evidence of absence. Therefore, in many cases, all that can be said is that at least exercise does not seem to make menopausal symptoms worse. I would recommend that the authors read, for example, Altman & Bland (1995) Statistics notes: Absence of evidence is not evidence of absence. BMJ 311: 485. doi: 10.1136/bmj.311.7003.485 and the first section in the recent blog by Frank Harrell https://www.fharrell.com/post/errmed/ and then go through their results.
Stating that there were no differences between groups is often problematic. For example, section 3.3. starts “The pooled effect size (SMD = 0.23, 95% CI: -0.1 to 0.56) showed no significant difference in general QoL of women in the exercise vs. the control groups.” I do not think we can say that either. My interpretation of the results would be something like “We could not confirm that exercise has beneficial effects on general (SMD = 0.23, 95% CI: -0.1 to 0.56), physical (), psychological () or social () QoL. However, it seems rather certain that exercise does not worsen QoL in women with menopausal symptoms”. Of course, I wish that the authors will make their own interpretations and express them in their writing style. Please also consider how you interpret results in sections 3.4. and 3.5. An example where the “no effect” seem justified is the case of yoga with general QoL (paragraph 3.5.). If you decide to interpret results differently, revise the discussion and conclusions.
Specific comments
Page 1, line 19: We found some improvements?
Page 1, line 25: PFMT non-significantly improved general QoL, what does that mean?
(Otherwise, I think the abstract flows now nicely.)
Page 2, line 73: I am sorry if I was unclear earlier; perhaps state that the study was reported according to the PRISMA guidelines as the recommendations go beyond search strategy.
Page 4, line 127: I am pleased that the authors took the time to use the ROB 2.0 tool. For clarity, perhaps change “revised ROB tool” to “ROB 2.0 tool.”
Page 5, line 163: “considered to indicate low publication bias”?
Page 7, lines 211–212: I still have concerns about this sentence, which now somewhat contradicts your meta-analysis results. Also, this is the issue by including Mercier et al. in the qualitative analysis – the study is not comparable to the RCTs. It should not be treated to provide a similar level of evidence as to the other two studies. It can be of course used in the discussion to strengthen the case of possible PFMT effects as the meta-analysis results were inconclusive.
Page 8: You have made excellent additions to table 2.
Author Response
The authors have clearly put in a lot of effort to improve the manuscript, and I thank them for their detailed responses. I think that the introduction and methods sections are ready; however, I would still recommend that the authors check how they interpret their results.
Broad comments
Points 1–3:
Point 1: What I see in the results is uncertainty and wide confidence intervals, which originates from a small number of studies with small sample sizes. Interpreting results from low-powered analyses is difficult. However, absence of evidence is not evidence of absence. Therefore, in many cases, all that can be said is that at least exercise does not seem to make menopausal symptoms worse. I would recommend that the authors read, for example, Altman & Bland (1995) Statistics notes: Absence of evidence is not evidence of absence. BMJ 311: 485. doi: 10.1136/bmj.311.7003.485 and the first section in the recent blog by Frank Harrell https://www.fharrell.com/post/errmed/ and then go through their results.
Stating that there were no differences between groups is often problematic. For example, section 3.3. starts “The pooled effect size (SMD = 0.23, 95% CI: -0.1 to 0.56) showed no significant difference in general QoL of women in the exercise vs. the control groups.” I do not think we can say that either. My interpretation of the results would be something like “We could not confirm that exercise has beneficial effects on general (SMD = 0.23, 95% CI: -0.1 to 0.56), physical (), psychological () or social () QoL. However, it seems rather certain that exercise does not worsen QoL in women with menopausal symptoms”. Of course, I wish that the authors will make their own interpretations and express them in their writing style. Please also consider how you interpret results in sections 3.4. and 3.5. An example where the “no effect” seem justified is the case of yoga with general QoL (paragraph 3.5.). If you decide to interpret results differently, revise the discussion and conclusions.
Point 2: Page 1, line 19: We found some improvements?
Point 3: Page 1, line 25: PFMT non-significantly improved general QoL, what does that mean?
(Otherwise, I think the abstract flows now nicely.)
Responses 1–3: Thanks for your valuable comments. We reinterpreted several findings in section 3.3 and 3.5 as you suggested. Along with this, we also edited related parts in the abstract, discussion, and conclusion as follows;
(In section 3.3)
Before revision> The pooled effect size (SMD = 0.23, 95% CI: -0.1 to 0.56) showed no significant difference in general QoL of women in the exercise vs. the control groups.
After revision> The pooled effect of exercise on general QoL was positive with small effect size, but the relation was not statistically significant (SMD = 0.23, 95% CI: -0.1 to 0.56).
(In section 3.5)
Before revision> Yoga appeared not to affect general QoL (SMD = -0.07, 95% CI: -0.46 to 0.33) but positively affected physical, psychological, sexual, and vasomotor symptoms.
After revision> In this review, we could not confirm the effect of yoga on general QoL (SMD = -0.07, 95% CI: -0.46 to 0.33). However, there were evidences for positive associations between yoga and physical, psychological, sexual, and vasomotor symptoms QoL scores.
(In the abstract)
Before revision> We found some improvements in physical and psychological QoL, but no significant improvements in general, social, and menopause-specific QoL in women with menopausal symptoms who exercised compared with those in the control groups. The most common interventions for women with menopausal and urinary symptoms were yoga and pelvic floor muscle training (PFMT), respectively. Although yoga positively affected physical QoL, its effects on general, psychological, sexual, and vasomotor symptoms QoL were inconclusive. PFMT non-significantly improved general QoL. Our findings suggest that well-designed studies are needed to confirm the effects of exercise on QoL in women with menopausal symptoms.
After revision> We found evidences for positive effects of exercise on physical and psychological QoL scores in women with menopausal symptoms. However, there was no evidence for the effects of exercise on general, social, and menopause-specific QoL scores. The most common interventions for women with menopausal and urinary symptoms were yoga and pelvic floor muscle training (PFMT), respectively. In our meta-analyses, while yoga significantly improved physical QoL, its effects on general, psychological, sexual, and vasomotor symptoms QoL scores as well as the effect of PFMT on general QoL were not significant. Our findings suggest that well-designed studies are needed to confirm the effect of exercise on QoL in women with menopausal symptoms.
(In the discussion)
Before revision> The present study found some improvements in physical and psychological QoL scores in women with menopausal symptoms after exercise interventions compared with no active interventions. However, there was no evidence for significant differences between the exercise and the control groups in terms of general, social, and menopause-specific QoL scores. Yoga and PFMT were respectively the most common interventions for women with menopausal and urinary symptoms in the RCTs included in this review. While yoga significantly improved physical QoL, its effects on general, psychological, sexual, and vasomotor symptoms QoL as well as the effects of PFMT on general QoL were non-significant.
After revision> The present study provided evidences for positive effects of exercise on physical and psychological QoL scores in women with menopausal symptoms. However, there was no evidence for the effects of exercise on general, social, and menopause-specific QoL scores. Yoga and PFMT, respectively, were the most common interventions for women with menopausal and urinary symptoms in the studies included in this review. In our meta-analyses, while yoga significantly improved physical QoL, its effects on general, psychological, sexual, and vasomotor symptoms QoL scores as well as the effect of PFMT on general QoL were not significant.
(In the conclusion)
Before revision> This systematic review found some improvements in physical and psychological QoL scores, but no significant improvements in general, social, and menopause-specific QoL scores, in women with menopausal symptoms after exercise interventions compared with no active interventions. Yoga and PFMT were respectively the most common interventions used for women with menopausal and urinary symptoms in the included studies. Although yoga significantly improved physical QoL, its effects on general, psychological, sexual, and vasomotor symptoms QoL were inconclusive. PFMT non-significantly improved general QoL. Our findings suggest that well-designed trials are needed to confirm the effects of exercise on QoL in women with menopausal symptoms.
After revision> This systematic review found positive effects of exercise on physical and psychological QoL scores in women with menopausal symptoms. However, there were no evident effects of exercise on general, social, and menopause-specific QoL scores, in women after exercise interventions compared with no active interventions. Yoga and PFMT, respectively, were the most common interventions for women with menopausal and urinary symptoms in the included studies. In our meta-analyses, while yoga significantly improved physical QoL, its effects on general, psychological, sexual, and vasomotor symptoms QoL scores as well as the effect of PFMT on general QoL were not significant. Our findings suggest that well-designed trials are needed to confirm the effect of exercise on QoL in women with menopausal symptoms.
Specific comments
Point 4: Page 2, line 73: I am sorry if I was unclear earlier; perhaps state that the study was reported according to the PRISMA guidelines as the recommendations go beyond search strategy.
Response 4: We described the sentence you pointed out before the search strategy section with some edited as follows; This study was reported following the PRISMA guidelines [33].
Point 5: Page 4, line 127: I am pleased that the authors took the time to use the ROB 2.0 tool. For clarity, perhaps change “revised ROB tool” to “ROB 2.0 tool.”
Response 5: We changed “revised ROB tool” to “ROB 2.0 tool” for clarity as you indicated.
In addition to this, we edited the name of this tool in the abstract as follows; We assessed the risk-of-bias in the included studies using the Cochrane risk-of-bias 2.0 tool for RCTs and computed converged standardized mean difference with 95% confidence interval.
Point 6: Page 5, line 163: “considered to indicate low publication bias”?
Response 6: Thank you for pointing it out correctly. We corrected this sentence as follows; Roughly symmetrical funnel plots were considered to indicate a low publication bias [54].
Point 7: Page 7, lines 211–212: I still have concerns about this sentence, which now somewhat contradicts your meta-analysis results. Also, this is the issue by including Mercier et al. in the qualitative analysis – the study is not comparable to the RCTs. It should not be treated to provide a similar level of evidence as to the other two studies. It can be of course used in the discussion to strengthen the case of possible PFMT effects as the meta-analysis results were inconclusive.
Response 7: Thank you for the suggestion.
Regarding your concern about the contradiction, effect sizes of the studies by Bertotto et al. and by Pereira et al. in terms of general and urinary symptoms QoL were 1.37 (95% CI: 0.55 to 2.19) and -1.43 (95% CI: -2.25 to -0.62), respectively. Both indicated a statistically significant improvement in these QoL domains in women with urinary symptoms. The effect size of the study by Pereira et al. in the general QoL analysis (SMD = 0.19, 95% CI: -0.53 to 0.90) was small but positive, indicating that there was still an improvement in general QoL in women with urinary symptoms although it was small and non-statistically significant. In conclusion, these findings were in line with the qualitative findings mentioned in the sentence.
Regarding your concern about including the study by Mercier et al. in the qualitative analysis, we explained the reason why we did so in the discussion as well as clarified the study design when describing the main findings of the included studies relating to the effects of PFMT as follows;
(In section 3.2)
Before revision> Regarding the main findings of the included studies, three studies on PFMT consistently showed increases in QoL [24,25,29], two of which were statistically significant [24,25] (Table 2).
After revision> Regarding the main findings of the included studies, three studies investigating the association between PFMT and QoL in women with urinary symptoms found several positive results [24,25,29]. While the single-arm study by Mercier et al. [24] and the RCT by Bertotto et al. [25] consistently showed statistically significant associations, Pereira et al. [29] showed a statistically significant large effect of PFMT on urinary symptoms and a statistically non-significant small effect of PFMT on general QoL.
(In discussion)
Before revision> Regarding different types of exercise interventions, similar to the main findings of three studies using PFMT included in this review [24,25,29], Cacciari et al. [68] indicated that PFMT can reduce the number of urinary leakage episodes and improve all types of urinary incontinence issues as well as QoL in women with urinary symptoms.
After revision> Regarding different types of exercise interventions, there were three studies [24,25,29] investigating the effect of PFMT on QoL in women with urinary symptoms included in this review. Although the study by Mercier et al. [24] was a single-arm study, it was not excluded from the qualitative synthesis because this study was a substudy of a RCT [68] which included three parallel evaluations at pre-intervention, 1-2 weeks after interventions, and a year post randomization, respectively. Data reported in the included study [24] were extracted from the first two evaluations. As the mother study [68] was strictly designed as a RCT, the study by Mercier et al. [24] was considered to meet the related selection criterion in this review. Furthermore, because the aims of the qualitative analysis were to compare and to assess the included studies in terms of methodological characteristics, interventions used, and reported findings, we included this study to provide a broad overview of the association between PFMT and QoL in women with urinary symptoms. Similar to the main findings of three included studies using PFMT, Cacciari et al. [69] indicated that PFMT can reduce the number of urinary leakage episodes and improve all types of urinary incontinence issues as well as QoL in women with urinary symptoms.
Point 8: Page 8: You have made excellent additions to table 2.
Response 8: Your comments were all valuable and very helpful for revising and improving our paper. Thanks again for your valuable comments.